# Divergent Copies of a *Cryptosporidium parvum*-Specific Subtelomeric Gene

**DOI:** 10.3390/microorganisms7090366

**Published:** 2019-09-18

**Authors:** Shijing Zhang, Li Chen, Falei Li, Na Li, Yaoyu Feng, Lihua Xiao

**Affiliations:** 1State Key Laboratory of Bioreactor Engineering, School of Resources and Environmental Engineering, East China University of Science and Technology, Shanghai 200237, China; zhangshijing1@foxmail.com (S.Z.); casper_chenli@126.com (L.C.); 2Key Laboratory of Zoonosis of Ministry of Agriculture, College of Veterinary Medicine, South China Agricultural University, Guangzhou 510642, China; falei0316@126.com (F.L.); nli@scau.edu.cn (N.L.)

**Keywords:** *Cryptosporidium parvum*, *cgd6_5520-5510* gene, subtype families, host range, genetic recombination

## Abstract

Subtype families of *Cryptosporidium parvum* differ in host range, with IIa and IId being found in a broad range of animals, IIc in humans, and IIo and IIp in some rodents. Previous studies indicated that the subtelomeric *cgd6_5520-5510* gene in *C. parvum* is lost in many *Cryptosporidium* species, and could potentially contribute to the broad host range of the former. In this study, we identified the presence of a second copy of the gene in some *C. parvum* subtype families with a broad host range, and showed sequence differences among them. The sequence differences in the *cgd6_5520-5510* gene were not segregated by the sequence type of the 60 kDa glycoprotein gene. Genetic recombination appeared to have played a role in generating divergent nucleotide sequences between copies and among subtype families. These data support the previous conclusion on the potential involvement of the insulinase-like protease encoded by the subtelomeric *cgd6_5520-5510* gene in the broad host range of *C. parvum* IIa and IId subtypes.

## 1. Introduction

*Cryptosporidium* spp. are common enteric pathogens, causing moderate to severe diarrhea in both humans and animals [1]. To date, about 40 *Cryptosporidium* species and approximately the same number of genotypes are recognized [2]. Most of them, such as *Cryptosporidium hominis* in humans and *Cryptosporidium tyzzeri* in house mice, have host specificity, being found in specific groups of animals. Among the few *Cryptosporidium* species with a broad host range, *Cryptosporidium parvum* is the most important zoonotic species, even though it is genetically related to *C. hominis* and *C. tyzzeri* [3,4]. Subtyping analysis based on the 60 kDa glycoprotein gene (*gp60*) has identified over 20 *C. parvum* subtype families [2]. Among the common ones, IIa appears to have the ability of infecting a broad range of animals as well as humans, while IIc appears to preferentially infect humans [2]. The story is complicated with the IId subtype family; while the IId subtype family is mostly found in sheep and goats in southern Europe, it is found in a broad range of animals in China, especially cattle [5]. Cattle in Egypt and Sweden are also frequently infected with the IId subtype family [2].

The genetic determinant of host adaptation in *C. parvum* subtypes remains unclear. In other apicomplexan parasites, gene duplication has been known as a major mechanism for adaptation to environmental changes [6,7,8]. Comparative genomics analysis of *C. parvum* and *C. hominis* has revealed the existence of copy-number variations in subtelomeric multigene families encoding the *Cryptosporidium*-specific MEDLE family of secreted proteins and insulinase-like proteases [3], suggesting that proteins encoded by these genes could be involved in host specificity of *Cryptosporidium* spp. Among them, *cgd6_5510* and *cgd6_5520* in chromosome 6, which encode insulinase-like proteases, are lost in many *Cryptosporidium* species such as *C. hominis* and *Cryptosporidium ubiquitum* [3,9]. There are also some variations in the number of subtelomeric genes of insulinase-like proteases between IIa and IId subtype families of *C. parvum* [10]. In addition, the *cgd6_5510* gene was previously shown to be highly polymorphic within *C. parvum* [11]. Recently, we have provided evidence indicating that *cgd6_5510* and *cgd6_5520* are actually misannotated fragments of the full gene ‘*cgd6_5520-5510*’ and the protein it encodes could be involved in the invasion of host cells by sporozoites [12].

Thus far, the sequence characteristics of the *cgd6_5520-5510* gene in divergent *C. parvum* subtypes are not clear. In our efforts of obtaining additional sequences of this gene from several subtype families of *C. parvum*, we have found the existence of multiple copies of the gene in some *C. parvum* subtype families.

## 2. Materials and Methods

### 2.1. Ethics Approval Statement

This study was approved by the Research Ethics Committee of the East China University of Science and Technology, with the approval number of 2015018 on 3rd Jan. 2015. Permission for collecting fecal specimens was obtained from owners or managers of the farms. Animals were handled in accordance with the Animal Ethics Procedures and Guidelines of the People’s Republic of China.

### 2.2. Cryptosporidium parvum Isolates

*C. parvum* IOWA isolate (subtype family IIa) oocysts were purchased from Waterborne, Inc. (New Orleans, LA, USA). Two fecal specimens (22578, and 27504) of the *C. parvum* IIdA19G1 subtype were collected from cattle in Shanghai and Henan, China. The presence of only a single subtype in IIa and IId isolates used was verified by examinations of the results of mapping of sequence reads from a separate study on the comparative genomics of *C. parvum* at the *gp60* and several other polymorphic loci. Other *C. parvum* isolates used included one fecal specimen (43849) of the IIcA5G3a subtype from a child in Egypt, four fecal specimens (25115, 25118, 25121, and 25125) of the IIoA14G1 subtype from *Macaca fascicularis* in Hainan, China, and four fecal specimens (27505, 27506, 27507, and 27508) of the IIpA9 subtype from bamboo rats in Hainan and Jiangxi, China. The *C. parvum* subtypes were identified by sequence analysis of the *gp60* gene [13]. Duplicate PCR was used in the analysis of each specimen, with DNA of *C. hominis* IbA10G2 subtype as the positive control. A negative control with sterile water was used in each PCR analysis. For the *C. parvum* IOWA (IIaA15G2R1) isolate, genomic DNA was isolated from 1 × 10^7^ oocysts using the QIAamp DNA Mini Kit (Qiagen, Hilden, Germany). For other isolates, genomic DNA was extracted from washed fecal specimens by using the FastDNA SPIN Kit for Soil (MP Biomedicals, Santa Ana, CA, USA). The genomic DNA was stored at –20 °C until PCR analysis.

### 2.3. PCR Analysis of cgd6_5520-5510

All PCR analyses in this study were performed with the KOD FX DNA Polymerase (TOYOBO, Osaka, Japan) in a total volume of 50 μL, containing 1 μL of DNA, 0.3 μM primers, 0.4 mM deoxyribonucleotide triphosphate, 25 μL of the 2 × KOD FX PCR buffer, and 1.0 U KOD FX DNA Polymerase. Whenever possible, single-round PCR was used in the amplification of the *cgd6_5520-5510* gene. As specimens of the *C. parvum* IIo and IIp subtypes had low oocyst counts, they were analyzed by nested PCR. For nested PCR, 2 μL of the primary PCR product was used in the secondary PCR. A negative control with sterile water was used in each PCR analysis. Primers used for the amplification of the *cgd6_5520-5510* gene in *C. parvum* IIa, IId, IIc, IIo, and IIp subtypes are listed in Table 1; Table 2. The PCR amplification was conducted under the following conditions: Denaturation at 95 °C for 5 min; 35 cycles of amplifications at 95 °C for 45 s, the specified annealing temperature for each primer set (Table 1 and Table 2) for 45 s, and 72 °C for 2 min (30 s for amplicons less than 1000 bp); and a final extension at 72 °C for 7 min. For low-concentration PCR products, we increased the number of PCR cycles for sequencing. PCR products of the *cgd6_5520-5510* gene generated were sequenced in both directions by Biosune Biotechnology, Shanghai, China. The data of PCR analyses were from at least three individual experiments.

### 2.4. TA Cloning of cgd6_5520-5510 PCR Products

To obtain sequences of both copies of the *cgd6_5520-5510* gene from isolates of the IIa and IId subtype families, PCR products of the expected size were excised from the agarose electrophoresis gel and purified by using the E.Z.N.A.^®^ Gel Extraction Kit (Omega bio-tek, Norcross, GA, USA). After incubation with DreamTaq PCR Master Mix (2 ×; Thermo Fisher Scientific, Waltham, MA, USA) at 72 °C for 30 min, the products were further purified using the E.Z.N.A.^®^ Cycle Pure Kit (Omega bio-tek, Norcross, GA, USA) and cloned into the pMD 18-T Vector (TaKaRa, Otsu, Shiga, Japan). *Escherichia coli* DH5α competent cells (Tiangen Biotech, Beijing, China) were transformed with the recombinant plasmid and grown in Luria-Bertani agar plates with 50 μg/mL ampicillin. Positive colonies were identified by PCR analysis and sequencing using universal primers M13F (5′-CGCCAGGGTTTTCCCAGTCACGAC-3′) and M13R (5′-AGCGGATAACAATTTCACACAGGA-3′) or specific primers designed for each subtype (Table 2). Positive clones were from at least three independent experiments of TA cloning (the *cgd6_5520-5510* gene was amplified from IIa or IId genomic DNA and cloned into the pMD 18-T Vector). We extracted four plasmids from four positive clones that have correct sequences of IIa-5520-5510-2, IId-5520-5510-1, IIo-5520-5510, and IIp-5520-5510, using E.Z.N.A.^®^ Plasmid Mini Kit I (Omega bio-tek, Norcross, GA, USA). The products were quantified by using the NanoDrop 2000 (Thermo Fisher Scientific, Waltham, MA, USA) and diluted 1:500 in sterile water. We used 1 μL of the corresponding product as the template in a 50 μL PCR reaction mixture.

### 2.5. Sequence Analysis

We identified additional *cgd6_5520-5510* gene sequences from the whole genome sequencing data in GenBank by using BLAST (https://blast.ncbi.nlm.nih.gov/Blast.cgi). Eighteen of these sequences are near full-length and used in the sequence alignment and phylogenetic analyses (Table 3). The nucleotide sequences of the *cgd6_5520-5510* gene obtained from *C. parvum* IIa, IId, IIc, IIo, and IIp subtypes were aligned with each other and the 18 sequences of the *cgd6_5520-5510* gene downloaded from the GenBank, using ClustalX version 2.0.11 (Http://www.clustal.org). A phylogenetic tree of the aligned sequences was constructed using the maximum likelihood method implemented in the MEGA 7 software (https://www.megasoftware.net/), based on the general time reversible model. The robustness of clad formation was assessed using bootstrap analysis with 1000 replicates. A sequence of the *cgd6_5520-5510* gene in *C. tyzzeri* (CTYZ_00001752 from CryptoDB database: https://cryptodb.org/cryptodb/) was used as the outgroup. Domains in the protein sequences encoded by the *cgd6_5520-5510* gene were identified by using HMMER (https://www.ebi.ac.uk/Tools/hmmer/search/hmmscan). The presence of divergent copies of the *cgd6_5520-5510* gene of the IOWA isolate was further confirmed by mapping of the Illumina sequence reads from a separate whole genome sequencing project to the reference IOWA genome in GenBank (AAEE00000000.1).

Fourteen *cgd6_5520-5510* gene sequences obtained in this study were submitted to the GenBank database under accession numbers MN090140 to MN090153.

## 3. Results

### 3.1. Presence of Mixed Sequences in cgd6_5520-5510 PCR Products from IIa and IId Subtypes

We initially amplified the 5′ and 3′ ends of the *cgd6_5520-5510* gene of *C. parvum* and sequenced the PCR products. In the sequence analysis of PCR products from the 5′ end of the gene in IId isolates, competing signals were present in the trace file after nucleotide position 528, leading to unreadable nucleotide sequences thereafter. This, however, was not seen in the DNA sequence analysis of IIa isolates (Figure 1A). Instead, there were 27 unresolved nucleotides presented in the ~818 bp fragment from the 3′ end of the *cgd6_5520-5510* gene in the IIa isolate. A total of 26 unresolved nucleotide were also seen in this region of the gene in IId isolates. An inspection of the trace files from the DNA sequencing identified the presence of two different nucleotides at these positions for both IIa and IId isolates. No unresolved nucleotides and underlying signals were observed in trace files from any PCR products from the 5′ end of the *cgd6_5520-5510* gene from IIa subtype, and from both ends of the gene of IIc, IIo, and IIp subtypes (Figure 1B).

### 3.2. Two Copies of cgd6_5520-5510 Gene in C. parvum IIa and IId Subtypes

We attempted to obtain sequences of the full or near full *cgd6_5520-5510* gene by sequencing TA-cloned PCR products. This led to the acquisition of the near complete *cgd6_5520-5510* gene from IIc, IIo, and IIp subtypes. Surprisingly, only one type of *cgd6_5520-5510* sequence was obtained from repeated sequencing of clones from the IIa or IId subtype. We obtained a total of 20 positive clones of the IIa-5520-5510-2 copy and 25 positive clones of the IId-5520-5510-1 copy from three individual experiments of TA cloning (Appendix A). For the IIa subtype, the sequence (IIa-5520-5510-2) obtained had 39 nucleotide differences and a 3 bp insertion compared with the *cgd6_5520-5510* sequence MK105815 (IIa-5520-5510-1) we previously obtained from the same IIa isolate [12]. Twenty-seven of these nucleotide substitutions occurred at the same positions of the ~818 bp fragment from the 3′ end of the *cgd6_5520-5510* gene analyzed above, indicating the presence of two copies of the *cgd6_5520-5510* gene in the IIa genome.

Based on the nucleotide sequence differences among IIa-5520-5510-1, IIa-5520-5510-2, and the *cgd6_5520-5510* sequence obtained above from the IId subtype, we designed copy-specific reverse primers (Specific-R1, Specific-R2; Appendix A). The use of the primers in PCR analysis of the *C. parvum* IOWA isolate yielded the expected copy-specific PCR products (Figure 2). DNA sequencing of the PCR products from genomic DNA generated sequences identical to IIa-5520-5510-1 and IIa-5520-5510-2. These PCR primers also amplified the two copies of the *cgd6_5520-5510* gene in the IId subtype (Figure 2), producing one sequence (IId-5520-5510-1) that was identical to the *cgd6_5520-5510* gene sequence obtained above from cloned IId PCR product, and another sequence (IId-5520-5510-2) that had 184 nucleotide substitutions and a 21 bp insertion. Twenty-six of these nucleotide substitutions occurred at the same positions of the ~818 bp fragment from the 3′ end of the *cgd6_5520-5510* gene analyzed above, indicating the presence of two copies of the *cgd6_5520-5510* gene in the IId genome as well. The amplification efficiency of each copy was different when using the same primer in TA cloning or using the copy-specific reverse primers designed at the same position of each copy (Figure 2 and Appendix A).

The amplification efficiency of IIa-5520-5510-1 was lower than that of the other copy in PCR analysis of the IIa subtype, while the efficiency of IId-5520-5510-1, which had the highest sequence identity (over 99%) to IIa-5520-5510-1, was higher than that of the other copy in the PCR analysis of the IId subtype. We also used these copy-specific primers to amplify the *cgd6_5520-5510* gene from genomic DNA of the IIc, IIo, and IIp subtypes, and obtained PCR products by one of the primer sets, which generated the same sequence we obtained above for each subtype.

Using this strategy, we obtained the full sequences of the IIa-5520-5510-2 type from IIa subtype family, IId-5520-5510-1 from the IId subtype family, and IIc-5520-5510 from the IIc subtype family. We also obtained near full sequences of the IIa-5520-5510-1 type from the IIa subtype family, IId-5520-5510-2 from the IId subtype family, IIo-5520-5510 from the IIo subtype family, and IIp-5520-5510 type from the IIp subtype family. A total of 15 *cgd6_5520-5510* sequences from 12 specimens were obtained in this study. As IIa-5520-5510-1 is identical to the published MK105815 in source and sequence, the remaining 14 *cgd6_5520-5510* gene sequences were submitted to the GenBank database under accession numbers MN090140 to MN090153.

### 3.3. Sequence Characteristics of cgd6_5520-5510 Gene from Different C. Parvum Subtype Families

There were ~1% sequence differences between the IIa-5520-5510-1 and IIa-5520-5510-2 types. All but one of the 39 nucleotide differences occurred within the 3′ end (*cgd6_5510* part) of the gene. By contrast, there were ~7% sequence differences between the two *cgd6_5520-5510* copies in the IId subtype, with nucleotide differences being present in both first and second halves of the full gene. Interestingly, there was over 99% nucleotide sequence identity between IIa-5520-5510-1 and IId-5520-5510-1 with the presence of only seven nucleotide differences over the full gene (Figure 3), while the differences between the other two types were relatively high (over 7%). The single-copied *cgd6_5520-5510* gene in IIc subtypes was also over 99% identical to IIa-5520-5510-1 type, with only 11 nucleotide differences and a 3 bp deletion. In particular, the sequence from IIcA5G3j had only three nucleotide differences compared with IIa-5520-5510-1. Similarly, the single-copied *cgd6_5520-5510* gene in the IIo and IIp subtypes had only six nucleotide differences between them. They, however, had 4%–7% sequence differences from the *cgd6_5520-5510* gene in IIa, IId, and IIc subtypes.

Using the BLAST analysis, we identified 26 *cgd6_5520-5510* gene sequences in GenBank (Table 3). The results of the BLAST analysis of the whole genome sequencing data showed that the two divergent copies of the *cgd6_5520-5510* gene were present in whole genome sequence data from at least four IIa isolates (IOWA, UKP1, UKP6, and 35090) and three IId isolates (UKP8, 31727, and 34902; Table 3). In all cases, these copies were present in two different contigs (under different GenBank accession numbers in Table 3), suggesting that the two copies of the *cgd6_5520-5510* gene were not presented in tandem. In isolate UKP1, which has fully assembled sequences at the chromosome level, the IIa-5520-5510-2 sequence was almost identical (with two nucleotide substitutions) to the 3363 bp sequence at the 5′ end of chromosome 6 (PYCJ01000002.1), while the partial IIa-5520-5510-1 sequence was identical to 901 bp at the 3′ end of chromosome 5 (PYCJ01000005.1), which has the remaining IIa-5520-5510-1 gene sequence missing. No other sequences of high identity were found in these genomes. The presence of two copies of the *cgd6_5520-5510* gene in the IOWA isolate was also confirmed by mapping of Illumina sequence reads we obtained from a separate whole genome sequencing project, which showed the presence of two types of sequence reads in the region of the *cgd6_5520-5510* gene (Appendix A).

Eighteen of these NCBI sequences are near full-length, thus were used in the alignment and phylogenetic analyses. In an alignment of these sequences and the 15 sequences of the *cgd6_5520-5510* gene from the present study, there were four major sequence types in the region of nucleotides 1 to 2607. Within the last 777 bp (nucleotides 2608 to 3384 in the alignment) of the gene, there were a total of 40 nucleotide substitutions, with mosaic distribution of the nucleotide substitutions among subtypes. Some of the 33 sequences were identical in the first 2607 bp, but were divergent in the remaining part of the gene (Figure 3 and Figure 4). For instance, nucleotide substitutions were detected in LKCL01000007.1, LKHM01000007.1, LKHN01000007.1, PYCJ01000002.1, LKHK02000007.1, BX538353.1, and IIa-5520-5510-2 sequences (all from IIa isolates) at positions 2618, 2662, and 2695 of the alignment. In contrast, IIc-5520-5510 has a sequence identical to IIa-5520-5510-1 and IId-5520-5510-1 in this region, despite having divergent sequences from them in the first 2607 bp, indicating the presence of mosaic sequence patterns (Figure 3 and Figure 4). The domain structure analysis revealed that all the proteins encoded by the *cgd6_5520-5510* gene had only two of the four domains in classic insulinases, and had the ‘HLLKQ’, ‘HLLEQ’, or ‘YLLEQ’ sequences instead of the Zn^2+^-binding motif ‘HXXEH’ in the active domain of insulinases (Figure 4 and Appendix A).

### 3.4. Phylogenetic Relationship Among cgd6_5520-5510 Gene Sequences

Phylogenetic analysis using the maximum likelihood method divided the *cgd6_5520-5510* sequences into two large groups, with discordant distribution of the sequences by *gp60* subtype (Figure 4). Among them, the two copies of the *cgd6_5520-5510* gene in IIa, IId-5520-5510-1 in IId and IIc-5520-5510 in IIc formed one large group, whereas the IId-5520-5510-2 in IId, IIo-5520-5510 in IIo, and IIp-5520-5510 in IIp formed the other group. By the *C. parvum gp60* subtype family, the two copies of the gene in the IIa subtype family were mostly in the first group. In contrast, the two copies of the gene in the IIaA15G1R1 subtype from Egypt were placed in different groups in the phylogenetic tree. The same was also true for the two copies of the gene in the IId subtype family. Similarly, the sequence of the gene in IIcA5G3j was placed in a group containing IIa-5520-5510-1, rather than together in a divergent clad containing sequences from other IIc isolates (Figure 4).

## 4. Discussion

Results of this study indicate the presence of two copies of the *cgd6_5520-5510* gene in the IIa and IId subtype families of *C. parvum*. Previous studies indicated that *cgd6_5520-5510* (MK105815, IIa-5520-5510-1 in this study) is a subtelomeric gene in the *C. parvum* genome and is lost in many other *Cryptosporidium* species [3,9]. In this study, we found competing signals and underlying peaks in the trace files from sequencing of the PCR products from the 5′ and 3′ ends of the *cgd6_5520-5510* gene in all *C. parvum* IIa and IId isolates, suggesting the presence of two copies of the *cgd6_5520-5510* gene. This was supported by a sequence analysis of cloned PCR products, in which we obtained a sequence of the *cgd6_5520-5510* gene from *C. parvum* IOWA that was different from the published one. The use of copy-specific primers in PCR had provided further support, generating two types of *cgd6_5520-5510* sequences from the isolate that differed by 1%. The long-term passage of the IOWA isolate in animal models and potential reproductive complexity of *C. parvum* probably have no significant effects on the presence of heterogeneous copies of the *cgd6_5520-5510* gene in *C. parvum* IIa isolates, as supported by the detection of them in other isolates (such as 35091, UKP1 and UKP6) in the analyses of whole genome sequence data (Table 3). Apparently, two divergent copies of the *cgd6_5520-5510* are also present in IId isolates, as revealed by analyses of DNA sequences from this study and whole genome projects in GenBank. In contrast, only one copy of the gene was found in the IIc, IIo, and IIp isolates in the analysis of sequences obtained from this study or GenBank. Therefore, there are apparent differences in the copy number of the *cgd6_5520-5510* gene among subtype families of *C. parvum*.

The location of the two copies of the *cgd6_5520-5510* gene in the *C. parvum* genome remains to be determined. The results of the BLAST analysis of whole genome sequencing data showed that the two copies from IIa or IId subtype were present in two different contigs, suggesting they are likely on two different chromosomes. This was supported by a bioinformatics analysis of whole genome sequence data from UKP1, which indicates that one copy of the gene (IIa-5520-5510-2) is located at the 5′ end of chromosome 6 and the other (IIa-5520-5510-1) is located at the 3′ end of chromosome 5. However, we found a discordance in the genomic location of the two copies of the *cgd6_5520-5510* gene in the two published genomes of chromosome 6 from the *C. parvum* IOWA isolate. In the BLAST analysis of NCBI sequences, IIa-5520-5510-1 was almost identical to the last two genes in chromosome 6 of the reference *C. parvum* IOWA genome: *cgd6_5520* (XM_001388322.1) and *cgd6_5510* (XM_625315.1), with the minor putative sequence differences attributable to the sequence errors present in the latter fragment of the reference genome [12]. It has, however, only 98.75% sequence identity to the last gene (1MB.841) in the chromosome 6 (BX538353.1) of the same isolate sequenced by another research group [14]. In contrast, IIa-5520-5510-2 was identical to the latter in the BLAST analysis, as also shown in the sequence alignment in the present study (Figure 3). Re-sequencing of the genome of the *C. parvum* IOWA isolate is needed to resolve this discrepancy in the genomic location of the two copies of the *cgd6_5520-5510* gene in the published genomes of this isolates.

In the sequence analysis, IIa-5520-5510-1, IId-5520-5510-1, and IIc-5520-5510 are highly similar in sequences, thus are probably orthologs in the *C. parvum* genome. Between the two copies of the *cgd6_5520-5510* gene in IId subtypes, IIa-5520-5510-1 is likely the ortholog of IId-5520-5510-1, as they have highly similar sequences. Thus, IIa-5520-5510-2 is likely the ortholog of IId-5520-5510-2, although there are ~7% sequence differences between them. The single-copied *cgd6_5520-5510* gene in the IIo and IIp subtype families are probably also orthologs to each other, as their sequences are similar. Although they are significantly different from other *cgd6_5520-5510* sequences, they are probably orthologs of IIa-5520-5510-1, IId-5520-5510-1, and IIc-5520-5510. Interestingly, although IIa-5520-5510-1 and IId-5520-5510-1 in *C. parvum* IIa and IId subtypes are highly identical in sequences, the amplification efficiency of each copy was different. The different amplification efficiency of each copy in the IIa or IId subtype could have resulted from the minor differences between nucleotide sequences, especially in the primer regions (Appendix A). We also used these copy-specific primers to amplify the *cgd6_5520-5510* gene from genomic DNA of the IIc, IIo, and IIp subtypes, and obtained PCR products by one of the primer sets. As Specific-R1 is more specific to IIc-5520-5510, and Specific-R2 is more specific to IIo-5520-5510 and IIp-5520-5510 in the primer region (Appendix A), the copy-specific PCR results of the IIc, IIo, and IIp subtypes are expected (Figure 2). Further studies are required to determine reasons for the differences in PCR amplification efficiency and to acquire the complete sequences of each gene copy in the genome of these divergent *C. parvum* subtypes.

The origin of the *cgd6_5520-5510* gene in divergent *C. parvum* subtype families is not clear. The single-copied gene in the IIc subtype is highly identical to one of the copies from the IIa or IId subtypes, indicating that they share a common ancestor. In contrast, the single-copied *cgd6_5520-5510* genes in the IIo and IIp subtypes are highly identical to each other but divergent from their orthologs in IIa, IIc, and IId subtypes. The second copy of the *cgd6_5520-5510* gene in IId subtypes, however, also has a very divergent sequence from the one shared by IIa, IIc, and IId subtypes. These divergent IId, IIo, and IIp sequences probably have a different origin.

The duplicated *cgd6_5520-5510* gene probably contributes to the broad host range of *C. parvum* IIa and IId subtypes. Comparative genomics analysis of *C. parvum* and *C. hominis* has revealed the existence of copy-number variations in subtelomeric multigene families encoding the *Cryptosporidium*-specific MEDLE family of secreted proteins and insulinase-like proteases [3], suggesting that proteins encoded by these genes could be involved in host specificity of *Cryptosporidium* spp. The *cgd6_5520-5510* gene, which encodes an insulinase-like protease, apparently has two copies in IIa and IId subtypes, but is a single-copied gene in IIc, IIo, and IIp subtypes. Among the *C. parvum* subtype families, IIa and IId are zoonotic subtypes with broader host ranges, while IIc preferentially infects humans, and IIo and IIp are adapted to some rodents [2,15]. The concordance between the host range or *C. parvum* subtypes and copy numbers of the *cgd6_5520-5510* gene suggests that the *cgd6_5520-5510* gene could contribute to the host specificity in *Cryptosporidium* spp., as previously indicated in comparative genomics analysis of *C. parvum* and *C. hominis* [3]. In apicomplexans, the genetic diversity resulted from copy number variations has been viewed as a major mechanism enabling parasites to adapt to new ecological changes [7,8]. Copy number variations, which result from gene duplication and natural selection, have significant impacts on gene expression and phenotypic variation [16,17].

Genetic recombination is apparently present in the *cgd6_5520-5510* gene of *C. parvum*. One of the two copies of the *cgd6_5520-5510* gene in the IId subtype family, IId-5520-5510-1, is virtually identical to one copy of the gene in IIa, IIa-5520-5510-1, except for the last 210 bp of the gene, which is identical to the other copy of the gene in IIa, IIa-5520-5510-2, in this region. This mosaic sequence pattern was also seen in the IIc subtype family. In fact, one of the IIc subtype, IIcA5G3j, has a sequence (PUXU01000016.1) of the *cgd6_5520-5510* gene almost identical to IIa-5520-5510-1, while the gene in other three IIc isolates has a more divergent sequence (Figure 3 and Figure 4). Similarly, one of the IIa isolates from Egypt, 35090 of the IIaA15G1R1 subtype, has a second copy of the gene with a sequence (LXLE01003238.1) that is identical to IId-5520-5510-2 in the IId subtype family. As both IIa and IId subtype families are common in bovine animals in Egypt [18,19], genetic recombination apparently has produced this discordance in subtyping between the *gp60* and *cgd6_5520-5510* loci. Genetic recombination has recently been identified as a mechanism for generating host-adapted and virulent *C. parvum* and *C. hominis* subtype families [3,20].

The transcription and the precise function of the *cgd6_5520-5510* gene are unclear. In our previous study, we analyzed the relative expression level of the *cgd6_5520-5510* gene over a 72-h time course in *C. parvum*-infected HCT-8 cells [12]. The expression of the *cgd6_5520-5510* gene was the highest at 2 h post-infection and declined thereafter, which was similar to the published data on the expression of the original *cgd6_5520* and *cgd6_5510* genes [21]. In other studies, copy number variations were associated with multiple orders of magnitude changes in gene expression [17,22,23]. The relative expression level of each *cgd6_5520-5510* copy in different subtypes remains to be determined. In this research, all the proteins encoded by the *cgd6_5520-5510* gene types are similar in domain structure. These proteins have only two of the four domains of classic insulinases, and the ‘HLLKQ’, ‘HLLEQ’, or ‘YLLEQ’ sequences instead of the core motif “HXXEH”. A complete insulinase such as the human insulinase (IDE_HUMAN, UniProtKB/Swiss-Prot: P14735.4) usually has four conserved domains; the N-terminal domain contains the inverted Zn^2+^-binding motif “HXXEH,” a core feature of M16 proteases, while the C-terminal domain is also required for dimerization and substrate recognition [24,25]. A previous study indicated that mutants with other motif sequences such as “HFCQH” have no proteolytic activities [26]. An earlier study showed that INS20-19 (encoded by IIa-5520-5510-1 in the present study) could be involved in the invasion or early developmental process of *C. parvum*, but is probably not a functional insulinase [12]. The divergent amino acid sequences at the core motif ‘HXXEH’ shown in the present study also supports the conclusion that these proteins are probably not functional insulinases. Further studies are needed to determine the precise function of the *cgd6_5520-5510* gene and investigate whether copy number variations of this gene in different *C. parvum* subtypes have an impact on gene expression.

## 5. Conclusions

This study has identified seven major sequence types of the *cgd6_5520-5510* gene among IIa, IId, IIc, IIo, and IIp subtype families of *C. parvum*, including two types each in the IIa and IId subtype families. These sequence types are discordant to sequence types at the *gp60* locus, with additional observation of mosaic sequence patterns among the *cgd6_5520-5510* sequence types. The likely presence of dual copies of the *cgd6_5520-5510* gene could contribute to the broad host range of *C. parvum* IIa and IId subtypes, while genetic recombination appears to be responsible for the high sequence diversity of the *cgd6_5520-5510* gene and the discordant subtyping results between the *cgd6_5520-5510* and *gp60* genes. These observations, however, need support from an extensive analysis of other isolates from these and other *C. parvum* subtype families, and direct whole-genome sequencing and comparative genomics analysis of various field isolates, including those included in the present study. In addition, careful re-sequencing, assembly, and re-annotation of the reference IOWA genome are needed to facilitate the identification of chromosomal location of these divergent copies in IIa and IId genomes. These approaches might lead to improved understanding of the genome structure and evolution of different *C. parvum* subtype families.

## Figures and Tables

**Figure 1 microorganisms-07-00366-f001:**
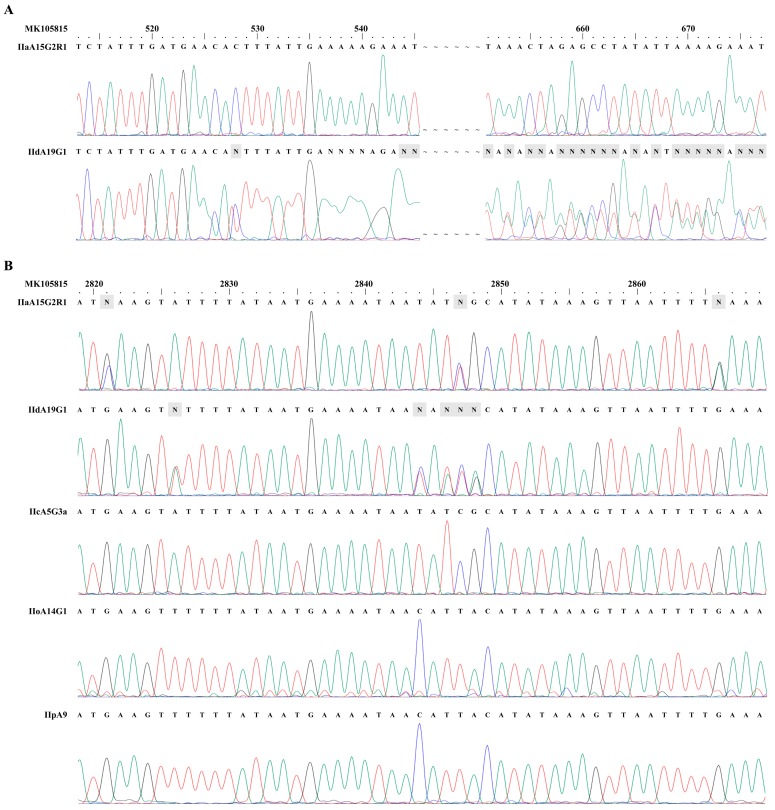
Presence of dual copies of the *cgd6_5520-5510* gene in *Cryptosporidium parvum* IIa and IId subtypes. (**A**) Sequencing trace files of PCR products of the 5′ end from the gene in *C. parvum* IIa and IId subtypes. Competing signals were present after nucleotide position 528 in the IId subtype. This, however, was not seen in the sequence analysis of the IIa isolates. (**B**) Sequencing trace files of PCR products of the 3′ end from the gene in *C. parvum* IIa, IId, IIc, IIo, and IIp subtypes. There were 27 unresolved nucleotides presented in the ~818 bp fragment from the 3′ end of the *cgd6_5520-5510* gene from the IIa subtype (three of which are shown as the letter N in the top panel in **B**) and 26 unresolved nucleotide from the IId subtype (five of which are shown as the letter N in the second panel in **B**). No unresolved nucleotides were present in trace files from IIc, IIo, and IIp subtype families. The reference sequence used in the sequence comparisons was MK105815. The peak maps in the trace files were exported by using ChromasPro version 1.41 (https://chromaspro.software.informer.com/1.4/). The adenine (A), guanine (G), cytosine (C) and thymine (T) peaks are green, black, blue and red, respectively.

**Figure 2 microorganisms-07-00366-f002:**
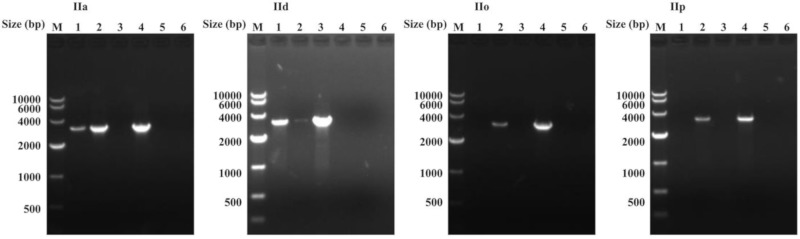
PCR amplification of the *cgd6_5520-5510* gene in *Cryptosporidium parvum* IIa, IId, IIo, and IIp genomic DNA using copy-specific primers using the corresponding plasmid from IIa-5520-5510-2, IId-5520-5510-1, IIo-5520-5510, or IIp-5520-5510 as a control for each subtype. The forward primer for all six lanes from the same subtype family is the same. Lane 1: PCR using Specific-R1 as the reverse primer; for IIa and IId subtypes, genomic DNA was used as the template, while for IIo and IIp subtypes, the primary PCR product was used as the template. Lane 2: PCR using Specific-R2 as the reverse primer; for IIa and IId subtype, genomic DNA was used as the template, while for IIo and IIp subtypes, the primary PCR products was used as the template. Lane 3: PCR using Specific-R1 as the reverse primer and the corresponding plasmid DNA for each subtype was used as the template. Lane 4: PCR using Specific-R2 as the reverse primer and the corresponding plasmid DNA for each subtype was used as the template; Lane 5: PCR using Specific-R1 as the reverse primer and sterile water was used as the template; Lane 6: PCR using Specific-R2 as the reverse primer and sterile water was used as the template. All these PCR products were sequenced. We amplified two copies of the *cgd6_5520-5510* gene from genomic DNA of the IIa and IId subtypes, but only one type of the *cgd6_5520-5510* gene from genomic DNA of the IIc (not shown), IIo, and IIp subtypes. Furthermore, we amplified PCR products from the corresponding plasmid of each subtype by only one of the primer sets, confirming the copy specificity of the primers. The results support the presence of two copies of the *cgd6_5520-5510* gene in *C. parvum* IIa and IId subtypes.

**Figure 3 microorganisms-07-00366-f003:**
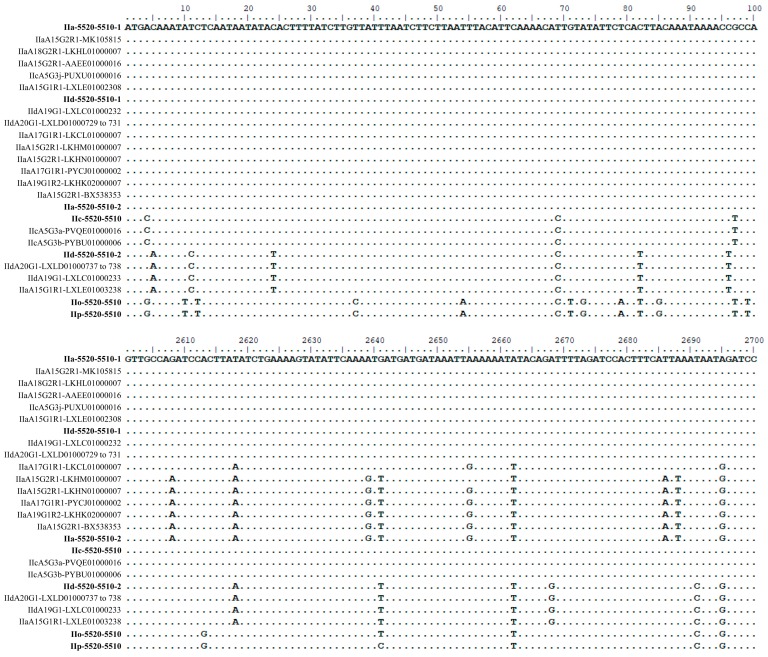
Alignment of unique nucleotide sequences of the *cgd6_5520-5510* gene obtained in this study together with 18 other *cgd6_5520-5510* sequences. Seven representative sequences of the *cgd6_5520-5510* gene obtained in this study are shown in bold. The remaining 18 sequences were downloaded from the GenBank database (Table 3). The nucleotides identical to the reference sequence (the first line of the alignment) are shown as dots. Some of the sequences, which are identical in the first 2607 bp, have a sequence similarity to others in the last 777 bp, leading to a mosaic sequence pattern at the 3′ end of the alignment.

**Figure 4 microorganisms-07-00366-f004:**
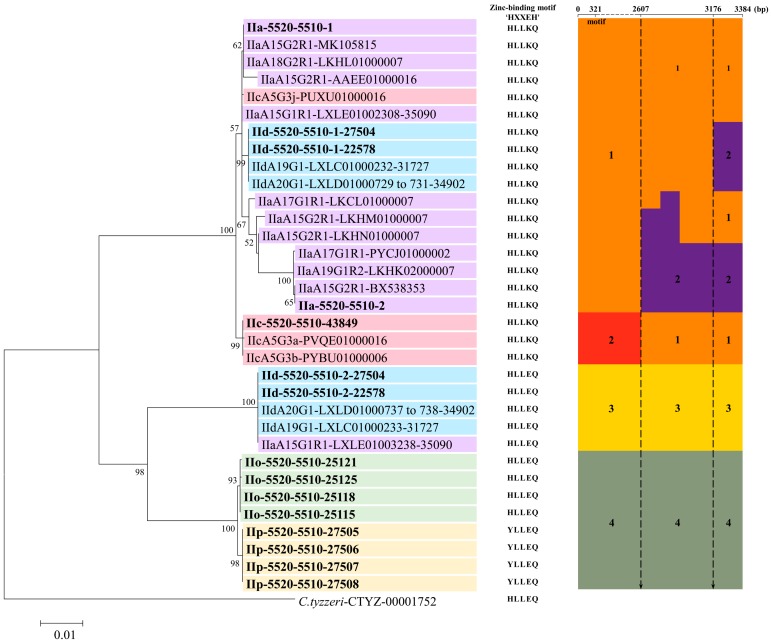
The phylogenetic relationship among 34 *cgd6_5520-5510* gene sequences. There is a discordance in subtyping between *cgd6_5520-5510* and *gp60* genes. The phylogenetic analysis based on the maximum likelihood method divided these *cgd6_5520-5510* gene types into two large groups. A sequence of the *cgd6_5520-5510* gene in *Cryptosporidium tyzzeri* (CTYZ_00001752 from CryptoDB database: https://cryptodb.org/cryptodb/) was used as the outgroup. Sequences from different subtype families are shown in different colors. The sequences obtained in this study are shown in bold. The zinc-binding motif of each sequence is shown in the middle of the figure. In the right of the figure, we have grouped these sequences by sequence similarity in different regions of the alignment. There are four major sequence types in the region of nucleotides 1 to 2607. Group 1 in orange was formed by sequences identical to IIa-5520-5510-1; group 2 in red was formed by sequences identical to IIc-5520-5510; group 3 in yellow was formed by sequences identical to IId-5520-5510-2; and group 4 in green was formed by sequences identical to IIo-5520-5510 or IIp-5520-5510. Within the last 777 bp, the three sequences from IIc subtypes are identical to IIa-5520-5510-1 and grouped together. There is a new group 2 in purple formed by sequences identical to IIa-5520-5510-2. The abscissa of the alignment is not drawn to scale.

**Table 1 microorganisms-07-00366-t001:** Primers designed to amplify the two copies of the *cgd6_5520-5510* gene in *Cryptosporidium parvum* IIa and IId subtypes.

Subtype Family	Target Region	Name of Primers	Forward Primer (5′ to 3′)	Annealing Temperature	PCR Product (bp)	Usage
			Reverse Primer (5′ to 3′)			
IIa, IId	5′ end of the *cgd6_5520-5510* gene	IId-F	CTTGTTATTTAATCTTCTTAATTTACATTC	52 °C	954–975	Found mixed sequences
		IId-R	ATAATAATAAGTTTTTCATTTCCAT			
	3′ end of the *cgd6_5520-5510* gene	IIa-F2	ACGCAAAAGTGATTAGTAGAATCTTTGATCTAGAC	52 °C	~818	
		IIa-R	CTCATCTTTCATGAATTTGAGTTGG			
	IIa-5520-5510-1, IId-5520-5510-1	IIa-F	ATGACAAATATCTCAATAATATACACTTTTATCTT	58 °C, 52 °C	~3165	Specific reverse primers
		Specific-R1	CTCCATAAATTTTTTAAAGGATGAATC			
	IIa-5520-5510-2, IId-5520-5510-2	IIa-F	ATGACAAATATCTCAATAATATACACTTTTATCTT	58 °C, 52 °C	~3171, ~3192	
		Specific-R2	CCACTCCATAAATTTATTAAAGGATAATAATTT			
IIc	IIc-5520-5510	IIa-F	ATGACAAATATCTCAATAATATACACTTTTATCTT	58 °C	~3162	Specific reverse primers
		Specific-R1	CTCCATAAATTTTTTAAAGGATGAATC			
	-	IIa-F	ATGACAAATATCTCAATAATATACACTTTTATCTT	-	-	
		Specific-R2	CCACTCCATAAATTTATTAAAGGATAATAATTT			
IIo, IIp	The full *cgd6_5520-5510* gene	IIa-F ^1^	ATGACAAATATCTCAATAATATACACTTTTATCTT	58 °C	~3374	Primary PCR
		IIa-R-new ^1^	TTCATGAATTTGAGTTGGCTATA			
	-	IIo-F^2^	AACCTCTAACACGCAAGATTACATA	-	-	Specific reverse primers
		Specific-R1 ^2^	CTCCATAAATTTTTTAAAGGATGAATC			
	IIo-5520-5510, IIp-5520-5510	IIo-F^2^	AACCTCTAACACGCAAGATTACATA	58 °C	~3192	
		Specific-R2 ^2^	CCACTCCATAAATTTATTAAAGGATAATAATTT			

^1^ Used as primary primers in nested PCR for IIo and IIp samples; ^2^ used as secondary primers in nested PCR for IIo and IIp samples.

**Table 2 microorganisms-07-00366-t002:** Primers used to amplify near full sequences of the *cgd6_5520-5510* gene in divergent *Cryptosporidium parvum* subtypes.

Subtype Family	Target Region	Name of Primers	Forward Primer (5′ to 3′)	Annealing Temperature	PCR Product (bp)	Usage
			Reverse Primer (5′ to 3′)			
IIa, IId	IIa-5520-5510-2, IId-5520-5510-1 full gene	IIa-F	ATGACAAATATCTCAATAATATACACTTTTATCTT	52 °C	~3360, ~3357	TA cloning
		IIa-R	CTCATCTTTCATGAATTTGAGTTGG			
	IIa-5520-5510-1 near full gene	IIa-F	ATGACAAATATCTCAATAATATACACTTTTATCTT	58 °C	~3165	Specific reverse primers
		Specific-R1	CTCCATAAATTTTTTAAAGGATGAATC			
	IId-5520-5510-2 near full gene	IIa-F	ATGACAAATATCTCAATAATATACACTTTTATCTT	52 °C	~3192	Specific reverse primers
		Specific-R2	CCACTCCATAAATTTATTAAAGGATAATAATTT			
IIc	IIc-5520-5510 full gene	IIa-F	ATGACAAATATCTCAATAATATACACTTTTATCTT	52 °C	~3354	Sequencing directly, TA cloning
		IIa-R	CTCATCTTTCATGAATTTGAGTTGG			
IIo, IIp	IIo-5520-5510, IIp-5520-5510 near full gene	IIa-F ^1^	ATGACAAATATCTCAATAATATACACTTTTATCTT	58 °C	~3374	Sequencing directly, TA cloning
		IIa-R-new ^1^	TTCATGAATTTGAGTTGGCTATA			
		IIo-F ^2^	AACCTCTAACACGCAAGATTACATA	58 °C	~3224	
		IIo-R ^2^	TTGTATGATCTAACCTTGTAAAG			

^1^ Used as primary primers in nested PCR for IIo and IIp samples; ^2^ used as secondary primers in nested PCR for IIo and IIp samples.

**Table 3 microorganisms-07-00366-t003:** Twenty-six sequences of the *cgd6_5520-5510* gene in *Cryptosporidium parvum* downloaded from the GenBank database.

Subtype Family	Subtype	Isolate	Geographic Location	Sequence Name in This Study	GenBank Accession Number	Nucleotide Position
IIa	IIaA15G1R1	35090 ^1^	Egypt: El Beheira	IIaA15G1R1-LXLE01002308	LXLE01002308.1	1 to 2825
				IIaA15G1R1-LXLE01003238	LXLE01003238.1	8331 to 11714
	IIaA15G2R1	IOWA ^1^	-	IIaA15G2R1-AAEE01000016	AAEE01000016.1	23074 to 26443
		IOWA ^1^	-	IIaA15G2R1-BX538353	BX538353.1	271827 to 275189
		IOWA ^1^	-	IIaA15G2R1-MK105815	MK105815.2	1 to 3360
		UKP4	United Kingdom: England	IIaA15G2R1-LKHM01000007	LKHM01000007.1	24207 to 27566
		UKP5	United Kingdom	IIaA15G2R1-LKHN01000007	LKHN01000007.1	24155 to 27514
		UKP6 ^1^	United Kingdom	-	LKCK01000007.1 ^2^	19930 to 21805
				-	LKCK01000011.1 ^2^	1196691 to 1198512
	IIaA17G1R1	UKP1 ^1^	United Kingdom	IIaA17G1R1-PYCJ01000002	PYCJ01000002.1	4407 to 7769
				-	PYCJ01000005.1 ^2^	1084332 to 1085232
		UKP7	United Kingdom: England	IIaA17G1R1-LKCL01000007	LKCL01000007.1	23021 to 26687
	IIaA18G2R1	UKP3	United Kingdom: Wales	IIaA18G2R1-LKHL01000007	LKHL01000007.1	25874 to 29233
	IIaA19G1R2	UKP2	United Kingdom: England	IIaA19G1R2-LKHK02000007	LKHK02000007.1	16497 to 19859
IIc	IIcA5G3a	UKP13	United Kingdom: England	-	PVQD01000016.1 ^2^	5597 to 6803
		UKP14	United Kingdom: England	-	PUXT01000016.1 ^2^	12896 to 16195
		UKP15	United Kingdom: Wales	IIcA5G3a-PVQE01000016	PVQE01000016.1	23118 to 26751
	IIcA5G3b	TU114	Uganda	IIcA5G3b-PYBU01000006	PYBU01000006.1	2767 to 6123
	IIcA5G3j	UKP16	United Kingdom: Wales	IIcA5G3j-PUXU01000016	PUXU01000016.1	26498 to 29857
	IIcA5G3p	UKP12	United Kingdom: England	-	PUXS01000016.1 ^2^	5184 to 5487
IId	IIdA19G1	31727 ^1^	China: Henan	IIdA19G1-LXLC01000232	LXLC01000232.1	130844 to 134203
				IIdA19G1-LXLC01000233	LXLC01000233.1	36841 to 40158
	IIdA20G1	34902 ^1^	Egypt: Kafr El Sheikh	IIdA20G1-LXLD01000729 to 731 ^3^	LXLD01000729.1	11759 to 12186
					LXLD01000730.1	1 to 699
					LXLD01000731.1	1 to 2342
				IIdA20G1-LXLD01000737 to 738 ^3^	LXLD01000737.1	11998 to 12428
					LXLD01000738.1	1 to 2363
	IIdA22G1	UKP8 ^1^	United Kingdom: England	-	LKCJ01000007.1 ^2^	27295 to 28283
				-	LKCJ01000010.1 ^2^	670660 to 673451

^1^ There are two copies of the *cgd6_5520-5510* gene in genomic sequences from these isolates. ^2^ These partial sequences were not used in the alignment and phylogenetic analyses. ^3^ Sequences were assembled from two or more whole genome shotgun sequences.

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
