# Peer review of "Divergent Copies of a Cryptosporidium parvum-Specific Subtelomeric Gene"

_microorganisms, 2019, doi:10.3390/microorganisms7090366_

Round 1

Reviewer 1 Report

The manuscript by Zhang and collaborators is a very good research work to understand the determinant of species-specificity of sub-types of the parasite Cryptosporidium parvum.

The Authors have carried out a thorough genetic characterization of the gene cgd6-5520-5510, which was previously thought as 2 separate coding regions. The comparison of the phylogeny based on this sequencing with the one based on the gp60 gene highlighted some differences, which could be used to better characterise the parasite sub-types and rationalise the host-specificity.

The authors have deposited the sequences in a public repository so their work on the sequences, SNPs and annotation might be readily incorporated into the data banks of this world-spread parasite.

Concerning the gene product, the Authors claim that it could be an insulinase-like protein, though they found some SNPs coding for amino acids substitutions in one of the proteins domain, namely the Zn finger one.

It would be helpful for a reader to have, at least in the supplementary material, a translation of the coding sequences aligned among them and also with a known insulinase-like protein, with boxes or highlights of the protein domains.

In this respect, the legend of Figure 4 should be reformulated to better explain the table on the far right, because it is not clear what are the colour codes referring to and what are the numbers 1 to 4 indicating? Are they the SNPs? The sub-types? The number of amino acids substituted in the consensus sequence? Please explain.

Other than that, the manuscript is well written and comprehensible. It opens the way to a better annotation of shotgun genomic sequences which is of utmost importance especially in the field of molecular parasitology.

—————

Do you have any ethical concerns about this study? NO (please check the right answer)

——————

Author Response

Thanks for the suggestions. We have addressed the issues raised:

We have added Figure S4 “A schematic representation of the domain structure of the cgd6_5520-5510 gene in parvum IIa, IId, IIc, IIo and IIp subtype families” in the supplementary materials. In addition, we have added the following sentences in discussion: “These proteins have only two of the four domains of classic insulinases, and the ‘HLLKQ’, ‘HLLEQ’ or ‘YLLEQ’ sequences instead of the core motif ‘HXXEH’. A complete insulinase such as the human insulinase (IDE_HUMAN) usually has four conserved domains. The N-terminal domain contains the inverted Zn2+-binding motif “HXXEH,” a core feature of all M16 proteases, while the C-terminal domain is also required for dimerization and substrate recognition [24,25]. A previous study indicated that mutants with other motif sequences such as “HFCQH” have no proteolytic activities [26].” (L423-432) We have changed the colors in Figure 4 and added the following sentences in its legend: “In the right of the figure, we have grouped these sequences by sequence similarity in different regions of the alignment. There are four major sequence types in the region of nucleotides 1 to 2607. Group 1 in orange was formed by sequences identical to IIa-5520-5510-1; Group 2 in red was formed by sequences identical to IIc-5520-5510; Group 3 in yellow was formed by sequences identical to IId-5520-5510-2; Group 4 in green was formed by sequences identical to IIo-5520-5510 or IIp-5520-5510. Within the last 777 bp, the three sequences from IIc subtypes are identical to IIa-5520-5510-1 and grouped together. There is a new group 2 in purple formed by sequences identical to IIa-5520-5510-2. The abscissa of the alignment is not drawn to scale.”

Reviewer 2 Report

Work of this type continues to contribute to understanding the characteristics of Cryptosporidium making it an exceedingly cosmopolitan pathogen.  The reproductive characteristics and virtually universal host range contribute to its highly adaptive success. Understanding the genetic basis for its pathogenicity is certainly of broad interest and the species provides an almost bottomless reservoir for investigation.

The description of finding dual copies in the IIa and IId DNA representatives investigated in this work may merit further elucidation. The source of IIa DNA, the Iowa Reference strain has a somewhat unique history, having been maintained by animal model passage since sometime in the decade of the 1980's. The potential effect of reproductive complexities of the species and this history on genetic details may be worth further investigation or at least comment. 

Author Response

We agree with the suggestion, and have added the following discussion on this issue to the first paragraph of the Discussion: “The long-term passage of the IOWA isolate in animal models and potential reproductive complexity of C. parvum probably have no significant effects on the presence of heterogeneous copies of the cgd6_5520-5510 gene in C. parvum IIa isolates, as supported by the detection of them in other isolates (such as 35091, UKP1 and UKP6) in the analyses of whole genome sequence data (Table 3).” Extensive changes and addition have been made at the suggestion of Reviewer 3 to increase the strength of the evidence provided (see tracked manuscript).

Reviewer 3 Report

A study on Cryptosporidium genetics, focusing on a subtelomeric sequence (cgd6_5520-5510) of interest for the biology of the microorganism, its genetics and genotyping. The results concern also epidemiological issues related to the typing of different species and their respective capability to infect different hosts. Even if most of the main conclusions require still additional experiments, the study contains information on the presence of divergent copies and allows several conclusions and speculations of interest for the scientific community in the field. The work is well conducted and approached by several appropriate methodologies.

However, some minor issues still may deserve further attention, e.g.:

- Line 168 et al: The question of the different amplification efficiency of the copies should be better supported in the experiments and discussed in the conclusions. In particular, the amplification has been performed also with the cloned sequences and not only with the whole extracted DNA? Can be excluded the presence of other similar sequences somewhere else along the genome? Since the genome is fully available, for several of the considered species, was any bioinformatics test performed or electronic pcr simulations? Since a main role is suggested to be played by Specific R1 and specific R2 reverse primers some additional information on their location and sequence differences should be added (e.g. a graphical schema to explain the hypothetical mechanism even just in supplemental materials). Was verified the presence of pure DNA from a single clone and excluded any possible contamination with other strains? How? Can be excluded a tandem amplification of the gene, or a redundancy at the same or different loci?

- Line 176 et al: the question of homologs and paralogs is interesting and sufficiently described but some data from whole genome sequencing should be added in supporting the proposed theories. Genome data are available, why were not used performing some bioinformatics tests?

- Line 192 etc: the effective meaning of the different SNPs should be analytically described to avoid confusions in the reader between an information aimed to support sequence differences and possible evolutionary/typing divergences and functional roles. In particular the question of the insulinases and/or Zn-binding domains is approached with an interesting biological description and speculation, but some data are available to support also a phenotypic and functional difference in C parvum?

- Line 249n etc.: The cgd6_5520-5510 gene from C parvum sequence that was different from the previously published one, was obtained from a single clone or how many? And out of a total of how many other obtained sequences? And of these how many were in accordance or not with this and/or the previously published ones? The total of the obtained data may be made available as supplementary material for the reader. In other words, could it be an artefact? Is it a constitutive mutation present in other strains or it could be considered as only of the studied clone? In the analysis of the different strains have been performed all controls to avoid a contamination of different species and/or their DNA..?

- Line 269 etc.: Yes, a whol genome study would be very useful, but any analysis was performed on the already available data accessing databases and using bioinformatics tools? Can this information on be added on the IOWA isolate or others before leaving the answer as an hypothesis for by the need of resequencing the genome and resolve the discrepancies in future experiments and further reports?

This question is fundamental to all the points concluded by the present paper, and in particular for the questions of SNPs, Paralogs, Othologs, observed differences in PCR amplification, etc..?

- Line 279 etc.: Amplification efficiency was tested using pure cloned sequences with the different primer under identical conditions, as controls? Was rigorously proved the absence of any contamination or pcr artefact? How?

- Line 284 etc.: again, before leaving to further studies the conclusion of the paragraph were done and/or can be performed some bioinformatics tests to confirm or exclude this hypothesis and/or provide additional hints to other experiments by the same or other authros?

- Line 293 etc. : Would not be possible to answer these issues, including the “”preferentially infec humans” question by analysis the available sequences and metagenomics data from other studies or available in databases e.g. NCBI? The association with infections and species seem overambitious and not supported by enough data, e.g. can other genes play a role (linked or not, paralog or included in totally independent loci and pathways? Are available other data on other mutations that could be considered in this context? And a comparison between different clones of the same species/strains was performed?).

- Line 303 etc.: Any data is available with RNA /gene expression and/or western or other onprotein analysis on the considered setting of samples and experiments?

- Line 324 etc.: and therefore, the available transcriptional data for cgd6_5520-5510 were considered even if limited? Why an RNA and/or Protein analysis was not performed on the different strains here reported? What about the statistics, to support or improve the strength of the conclusion (or hypothesis?) that “these proteins are probably not functional insulinases”? So, for this and previous comments the conclusion that “genetic recombination appears to be responsible” for the observed discrepancies is probably correct and interesting, but not clearly supported by consistent data.

Moreover, some information on the effective workload performed should be made available/transparent (in mat and met or in suppl materials), e.g. how many experiments were done for each set? All the strains were analysed for all the experiments in parallel? Additional information -if available- should be added, (in the text and/or in supplemental materials), and a paragraph on the limits of the study must be added reporting all critical considerations and possible major criticisms related to the general adopted strategy and on the possible gap between acquired data (including those from literature and already available databases) and conclusions, finally distinguishing demonstrated conclusions from enticing (even if definitely reasonable) hypothesis. Please consider to improve your manuscript in the light of these hints.

Author Response

Many thanks for the constructive comments and suggestions from the reviewer. We have addressed each of them in the revised manuscript. Below are the specifics of the revision, with suggestions/comments from the reviewer being listed first, followed by our responses.

- Line 168 et al: The question of the different amplification efficiency of the copies should be better supported in the experiments and discussed in the conclusions. In particular, the amplification has been performed also with the cloned sequences and not only with the whole extracted DNA? Can be excluded the presence of other similar sequences somewhere else along the genome? Since the genome is fully available, for several of the considered species, was any bioinformatics test performed or electronic pcr simulations? Since a main role is suggested to be played by Specific R1 and specific R2 reverse primers some additional information on their location and sequence differences should be added (e.g. a graphical schema to explain the hypothetical mechanism even just in supplemental materials). Was verified the presence of pure DNA from a single clone and excluded any possible contamination with other strains? How? Can be excluded a tandem amplification of the gene, or a redundancy at the same or different loci?

Response:

As we only have plasmids of one copy of the gene for each of the IIa and IId subtypes (the other copy in IIa or IId was obtained by direct PCR and sequence analysis using copy-specific primers), we could not perform a direct comparison of PCR amplification efficiency using plasmid DNA as suggested. As the reviewer appears to worry about potential PCR contamination during the analysis, we have repeated the PCR in the original Figure 2 by including plasmid DNA from our previous positive clones as controls. As a result, a new Figure 2 has replaced the previous figure with more robust data. As we can see, using Specific-R1 and Specific-R2, we have obtained similar amplification results for IIa, IId, IIo and IIp subtype families as in the previous figure. These primers only amplified plasmid DNA from their perspective copies. These new results provide stronger evidence, together with bioinformatics analysis of whole genome sequence data from GenBank, on the presence of two copies of the cgd6_5520-5510 gene in IIa and IId subtypes, but only one type in the other subtypes. Accordingly, additions have been made to the Materials and Methods (L113-118) and Figure 2 legend. In addition, we have removed the sentence “In PCR analysis of the cgd6_5520-5510 gene in the IIo and IIp subtypes, the amplification efficiency of Specific-R1 was significantly lower than that of Specific-R2. In contrast, the Specific-R1 amplified the cgd6_5520-5510 gene of the IIc subtype more effectively than Specific-R2.” (L226-228)

Regarding the possible presence of other similar sequences somewhere else along the genome, we have re-done the BLAST analysis of the GenBank data from C. parvum and updated Table 3 by including information on partial sequences of the cgd6_5520-5510 gene. In addition, we have added the following text to the Results: “The results of the BLAST analysis of the whole genome sequencing data showed that the two divergent copies of the cgd6_5520-5510 gene were present in whole genome sequence data from at least four IIa isolates (IOWA, UKP1, UKP6, and 35090) and three IId isolates (UKP8, 31727 and 34902) (Table 3). In all cases, these copies were present in two different contigs (under different GenBank accession numbers in Table 3), suggesting that the two copies of the cgd6_5520-5510 gene were not presented in tandem. In isolate UKP1, which has fully assembled sequences at the chromosome level, the IIa-5520-5510-2 sequence was almost identical (with two nucleotide substitutions) to the 3,363-bp sequence at the 5' end of chromosome 6 (PYCJ01000002.1), while the partial IIa-5520-5510-1 sequence was identical to 901-bp at the 3' end of chromosome 5 (PYCJ01000005.1), which has the remaining IIa-5520-5510-1 gene sequence missing. No other sequences of high identity were found in these genomes.” (L252-262).

The added information should have provided further evidence on the presence of two divergent copies of the cgd6_5520-5510 gene in other IIa and IId isolates sequenced by us others in previous whole genome sequencing projects. They have also shed light on their potential locations on two chromosomes (rather than next to each other in tandem on one chromosome).

 As suggested, we have added the Figure S2 “Location and sequence differences of the copy-specific reverse primers in the alignment of cgd6_5520-5510 sequences from C. parvum IIa, IId, IIc, IIo and IIp subtypes.” to explain the different amplification efficiency of the copies in this article. In addition, we have added the following sentences to the Discussion: “We also used these copy-specific primers to amplify the cgd6_5520-5510 gene from genomic DNA of the IIc, IIo and IIp subtypes, and obtained PCR products by one of the primer sets. As Specific-R1 is more specific for IIc-5520-5510, and Specific-R2 is more specific to IIo-5520-5510 and IIp-5520-5510 in the primer region (Figure S2), the copy-specific PCR results of the IIc, IIo and IIp subtypes are expected (Figure 2). Further studies are required to determine reasons for the differences in PCR amplification efficiency and to acquire the complete sequences of each gene copy in the genome of these divergent C. parvum subtypes.” (L370-377).

We verified the presence of pure DNA representing only a single isolate/strain by mapping of sequence reads to gp60 and several other polymorphic genes, as the IIa and IId isolates used in the study have been sequenced for the whole genome since the completion of present study. We have added the sentence to Materials and Methods: “The presence of only a single subtype in IIa and IId isolates used was verified by examinations of the results of mapping of sequence reads from a separate study on comparative genomics of C. parvum at the gp60 and several other polymorphic loci.” (L67-70).

- Line 176 et al: the question of homologs and paralogs is interesting and sufficiently described but some data from whole genome sequencing should be added in supporting the proposed theories. Genome data are available, why were not used performing some bioinformatics tests?

Response:

We have used BLAST analysis of whole genome sequence data in the GenBank to support the identification of homologs and paralogs of the cgd6_5520-5510 gene. As described above, the outcome of this analysis has been described in great detail in Results and Discussion. The GenBank sequences used in sequence alignment and phylogenetic analysis (Figures 3 and 4) were identified by this bioinformatics analysis. We have also done some mapping of Illumina sequence reads from the IOWA isolate (see below). More advanced bioinformatics analyses of published genomes probably require consent from data submitters and are beyond the scope of this work.

- Line 192 etc: the effective meaning of the different SNPs should be analytically described to avoid confusions in the reader between an information aimed to support sequence differences and possible evolutionary/typing divergences and functional roles. In particular the question of the insulinases and/or Zn-binding domains is approached with an interesting biological description and speculation, but some data are available to support also a phenotypic and functional difference in C parvum?

Response:

We have changed the term “SNPs” to “nucleotide differences” or “nucleotide substitutions” to avoid the confusions, as we did not conduct further analysis on the nature (synonymous versus nonsynonymous) of the nucleotide substitutions. We also have no experimental data on phenotypic differences in functions of the proteins encoded by different copies of the cgd6_5520-5510 gene in C. parvum.

- Line 249n etc.: The cgd6_5520-5510 gene from C parvum sequence that was different from the previously published one, was obtained from a single clone or how many? And out of a total of how many other obtained sequences? And of these how many were in accordance or not with this and/or the previously published ones? The total of the obtained data may be made available as supplementary material for the reader. In other words, could it be an artefact? Is it a constitutive mutation present in other strains or it could be considered as only of the studied clone? In the analysis of the different strains have been performed all controls to avoid a contamination of different species and/or their DNA..?

Response:

We have added the raw data from the TA cloning to “Figure S1, Table S1 and Table S2” in the supplementary materials. They provide details on the number of isolates and clones analyzed, their sequence identity to references, etc. As indicated in the updated Table3,both divergent copies of the gene have been seen in four IIa isolates and three IID isolates, and data generated in this study were from multiple clones of the IIa and IId isolates used. Therefore, we have added the following sentences to the Results: “We obtained a total of 20 positive clones of the IIa-5520-5510-2 copy and 25 positive clones of the IId-5520-5510-1 copy from three individual experiments of TA cloning (Figure S1, Table S1 and Table S2)” (L174-176)”, and “A total of 15 cgd6_5520-5510 sequences from 12 specimens were obtained in this study. As IIa-5520-5510-1 is identical to the published MK105815 in source and sequence, the remaining 14 cgd6_5520-5510 gene sequences were submitted to the GenBank database under accession numbers MN090140 to MN090153” (L233-236).

Some sequences from IIa or IId subtype families we generated in the study are identical to the published ones, although their subtype identity and sources are different. The sequences from IIo and IIp subtypes are novel and very different. As we also obtained two types of sequences from some of the IIa and IId isolates sequenced at the whole genome level by others (see Table 3 and Figure 4), they are unlikely artifacts generated by PCR or DNA sequencing in this study.

The sequences are not artefacts. The sequences from IIa and IId subtypes were verified by both TA cloning and copy-specific PCR, and the sequences from IIc, IIo and IIp subtypes were verified by both PCR and TA cloning. Most of mutations have been see in other isolates based on our own data and whole genome sequence data from GenBank.

In the Materials and Methods, we have added the following sentences: “A negative control with sterile water was used in each PCR analysis” (L88), “The data of PCR analyses were from at least three individual experiments” (L96), and “The sequencing data of TA cloning were from at least three independent experiments” (L113-114).

- Line 269 etc.: Yes, a whole genome study would be very useful, but any analysis was performed on the already available data accessing databases and using bioinformatics tools? Can this information on be added on the IOWA isolate or others before leaving the answer as an hypothesis for by the need of resequencing the genome and resolve the discrepancies in future experiments and further reports? This question is fundamental to all the points concluded by the present paper, and in particular for the questions of SNPs, Paralogs, Othologs, observed differences in PCR amplification, etc..?

Response:

Most of the analyses of whole genome sequence data in GenBank are BLAST-based, as described above. We did do some mapping of Illumina sequence reads of the IOWA isolate from a separate whole genome sequencing project to the C. parvum reference genome. The results of this analysis showed the presence of two types of reads mapped to the area of the cgd6_5520-5510 gene in chromosome 6. We have added Figure S3 to the revised report and the following sentence to the Results: “The presence of two copies of the cgd6_5520-5510 gene in the IOWA isolate was also confirmed by mapping of Illumina sequence reads we obtained from a separate whole genome sequencing project, which showed the presence of two types of sequence reads in the region of the cgd6_5520-5510 gene (Figure S3).” (L262-265)

- Line 279 etc.: Amplification efficiency was tested using pure cloned sequences with the different primer under identical conditions, as controls? Was rigorously proved the absence of any contamination or pcr artefact? How?

Response: We share the concern on PCR contamination. As described above, we have used plasmid DNA in our PCR in the work for the new Figure 2, and added statements on numbers of replicate analyses used in this study (see above). The same results were generated from all independent experiments, as shown in the new and old Figure 2 in the tracked manuscript.

- Line 284 etc.: again, before leaving to further studies the conclusion of the paragraph were done and/or can be performed some bioinformatics tests to confirm or exclude this hypothesis and/or provide additional hints to other experiments by the same or other authros?

Response:

As suggested, we have more robust bioinformatics analyses, thus expanded Table 3 with new results and added Figures S3 and S4 on outcome of these additional analyses. New descriptions on these data have been added to the M&M (L67-70, L120-123, L133-136), Results (L233-236, L251-266), and Discussion (L339-346,L423-434).

- Line 293 etc. : Would not be possible to answer these issues, including the “”preferentially infec humans” question by analysis the available sequences and metagenomics data from other studies or available in databases e.g. NCBI? The association with infections and species seem overambitious and not supported by enough data, e.g. can other genes play a role (linked or not, paralog or included in totally independent loci and pathways? Are available other data on other mutations that could be considered in this context? And a comparison between different clones of the same species/strains was performed?).

Response:

We agree with your comments as other apicomplexans have been known to use multiple strategies for invasion. We have changed the sentence from “The duplicated cgd6_5520-5510 gene could contribute to the broad host range of C. parvum IIa and IId subtypes.” to “The duplicated cgd6_5520-5510 gene probably contribute to the broad host range of C. parvum IIa and IId subtypes.” and we have added the sentences: “Comparative genomics analysis of C. parvum and C. hominis has revealed the existence of copy-number variations in subtelomeric multigene families encoding the Cryptosporidium-specific MEDLE family of secreted proteins and insulinase-like proteases [3], suggesting that proteins encoded by these genes could be involved in host specificity of Cryptosporidium spp.” In addition, we have added the following sentences to the end of the Conclusions: “ These observations, however, need support from extensive analysis of other isolates from these and other C. parvum subtype families, and direct whole-genome sequencing and comparative genomics analysis of various field isolates, including those included in the present study. In addition, careful re-sequencing, assembly, and re-annotation of the reference IOWA genome are needed to facilitate the identification of chromosomal location of these divergent copies in IIa and IId genomes. These approaches might lead to improved understanding of the genome structure and evolution of different C. parvum subtype families.”

- Line 303 etc.: Any data is available with RNA /gene expression and/or western or other on protein analysis on the considered setting of samples and experiments?

Response:

We have added the following sentences in discussion: “In our previous study, we analyzed the relative expression level of the cgd6_5520-5510 gene over a 72-h time course in C. parvum-infected HCT-8 cells [12]. The expression of the cgd6_5520-5510 gene was the highest at 2 h post-infection and declined thereafter, which was similar to the published data on the expression of the original cgd6_5520 and cgd6_5510 genes [21]” and “The relative expression level of each cgd6_5520-5510 copy in different subtypes remains to be determined”. We have further removed the sentence: “Thus, the transcriptional levels of the cgd6_5520-5510 gene in various C. parvum subtypes could be different.”

- Line 324 etc.: and therefore, the available transcriptional data for cgd6_5520-5510 were considered even if limited? Why an RNA and/or Protein analysis was not performed on the different strains here reported? What about the statistics, to support or improve the strength of the conclusion (or hypothesis?) that “these proteins are probably not functional insulinases”? So, for this and previous comments the conclusion that “genetic recombination appears to be responsible” for the observed discrepancies is probably correct and interesting, but not clearly supported by consistent data.

Response:

We have added the sentences: “In our previous study, we analyzed the relative expression level of the cgd6_5520-5510 gene over a 72-h time course in C. parvum-infected HCT-8 cells. The expression of the cgd6_5520-5510 gene was the highest at 2 h post-infection and declined thereafter, which was similar to the published data on the expression of the original cgd6_5520 and cgd6_5510 genes.” “The relative expression level of each cgd6_5520-5510 copy in different subtypes remains to be determined”.

We have added Figure S4 “A schematic representation of the domain structure of the cgd6_5520-5510 gene in C. parvum IIa, IId, IIc, IIo and IIp subtype families.” in supplementary materials. We have improved our analysis of insulinases and added the following sentences in discussion: “These proteins have only two of the four domains of classic insulinases, and have the ‘HLLKQ’, ‘HLLEQ’ or ‘YLLEQ’ sequences instead of the core motif ‘HXXEH’. A complete insulinase such as the human insulinase (IDE_HUMAN) usually has four conserved domains; the N-terminal domain contains the inverted Zn2+-binding motif “HXXEH,” a core feature of M16 proteases, while the C-terminal domain is also required for dimerization and substrate recognition [23,24]. A previous study indicated that mutants with other motif sequences such as “HFCQH” have no proteolytic activities [25].” (423-434)

-Moreover, some information on the effective workload performed should be made available/transparent (in mat and met or in suppl materials), e.g. how many experiments were done for each set? All the strains were analysed for all the experiments in parallel? Additional information -if available- should be added, (in the text and/or in supplemental materials), and a paragraph on the limits of the study must be added reporting all critical considerations and possible major criticisms related to the general adopted strategy and on the possible gap between acquired data (including those from literature and already available databases) and conclusions, finally distinguishing demonstrated conclusions from enticing (even if definitely reasonable) hypothesis. Please consider to improve your manuscript in the light of these hints.

Response:

Thanks for your comments. In the Materials and Methods, we have added the following sentences: “The data of PCR analyses were from at least three individual experiments”, “The sequencing data of TA cloning were from at least three independent experiments”, “We obtained a total of 20 positive clones of IIa-5520-5510-2 copy and 25 positive clones of IId-5520-5510-1 copy respectively from three individual experiments of TA cloning”, “A negative control with sterile water was used in each PCR analysis”, “For low-concentration PCR products, we then increased the number of PCR cycles for sequencing”.

We have provided detailed results of the cloning and sequencing of the PCR products in two supplementary tables and one supplemental figure. We have added the controls (pure cloned sequences) suggested by you and replaced the Figure 2 with a new one. We have modified some of the statements to differentiate data from our own analysis, data from bioinformatics analysis of GenBank data, and interpretations of published observations.

To put our observations/conclusions in perspective, as suggested, we have added the following sentences to the end of the Conclusions: “These observations, however, need support from extensive analysis of other isolates from these and other C. parvum subtype families, and direct whole-genome sequencing and comparative genomics analysis of various field isolates, including those included in the present study. In addition, careful re-sequencing, assembly, and re-annotation of the reference IOWA genome are needed to facilitate the identification of chromosomal location of these divergent copies in IIa and IId genomes. These approaches might lead to improved understanding of the genome structure and evolution of different C. parvum subtype families.”

Round 2

Reviewer 3 Report

All comments and concerns were clearly and rigorously fulfilled. Congratulation for your work and all the best.